# Enzyme Bioprospection of Marine-Derived Actinobacteria from the Chilean Coast and New Insight in the Mechanism of Keratin Degradation in *Streptomyces* sp. G11C

**DOI:** 10.3390/md18110537

**Published:** 2020-10-28

**Authors:** Valentina González, María José Vargas-Straube, Walter O. Beys-da-Silva, Lucélia Santi, Pedro Valencia, Fabrizio Beltrametti, Beatriz Cámara

**Affiliations:** 1Laboratorio de Microbiología Molecular y Biotecnología Ambiental, Departamento de Química y Centro de Biotecnología Daniel Alkalay Lowitt, Universidad Técnica Federico Santa María, Valparaíso 2340000, Chile; valentina.gonzalez.f10@gmail.com (V.G.); maru.vs@gmail.com (M.J.V.-S.); 2Faculdade de Farmácia, Universidade Federal do Rio Grande do Sul, Porto Alegre, RS 90610-000, Brazil; walter.beys@ufrgs.br (W.O.B.-d.-S.); lucelia.santi@ufrgs.br (L.S.); 3Laboratorio de Biocatálisis y Procesamiento de Alimentos, Departamento de Ingeniería Química y Ambiental, Universidad Técnica Federico Santa María, Valparaíso 2340000, Chile; pedro.valencia@usm.cl; 4BioC-CheM Solutions S.r.l., 21040 Gerenzano, Italy; fbeltrametti@BioC-CheMsolutions.com

**Keywords:** marine actinobacteria, *Streptomyces*, rare actinobacteria, hydrolytic enzymes, keratinolytic proteases, secretome

## Abstract

Marine actinobacteria are viewed as a promising source of enzymes with potential technological applications. They contribute to the turnover of complex biopolymers, such as pectin, lignocellulose, chitin, and keratin, being able to secrete a wide variety of extracellular enzymes. Among these, keratinases are a valuable alternative for recycling keratin-rich waste, which is generated in large quantities by the poultry industry. In this work, we explored the biocatalytic potential of 75 marine-derived actinobacterial strains, focusing mainly on the search for keratinases. A major part of the strains secreted industrially important enzymes, such as proteases, lipases, cellulases, amylases, and keratinases. Among these, we identified two streptomycete strains that presented great potential for recycling keratin wastes—*Streptomyces* sp. CHA1 and *Streptomyces* sp. G11C. Substrate concentration, incubation temperature, and, to a lesser extent, inoculum size were found to be important parameters that influenced the production of keratinolytic enzymes in both strains. In addition, proteomic analysis of culture broths from *Streptomyces* sp. G11C on turkey feathers showed a high abundance and diversity of peptidases, belonging mainly to the serine and metallo-superfamilies. Two proteases from families S08 and M06 were highly expressed. These results contributed to elucidate the mechanism of keratin degradation mediated by streptomycetes.

## 1. Introduction

Biocatalysis has become a relevant alternative to chemical processes, being recognized by many industries. However, the industrial applications of the enzymes have been hampered mainly owing to undesirable property in terms of stability, catalytic efficiency, and specificity [1]. To overcome such difficulties, one of the approaches used is the search for enzymes in novel natural niches, such as the oceans or other extreme environments [2,3]. The ocean contains a great reservoir of biodiversity. Salinity, low temperature, high pressure, oligotrophic conditions, widely ranging mineral content in seawater, and special lighting conditions can contribute to generating differences between the enzymes synthesized by marine and terrestrial microorganisms [4,5]. Actinobacteria constitute a rich source of novel biocatalysts for industrial utilization [2]. In nature, this group of Gram-positive bacteria contributes notably to the turnover of complex biopolymers, such as lignocellulose, pectin, keratin, and chitin, being able to synthesize a wide variety of enzymes [2,6]. In fact, a large number of reports of enzymes produced by actinobacteria have been described, such as proteases, keratinases, lipases, cellulases, chitinases, among others [7,8,9,10,11]. Nevertheless, limited literature is available on the diversity and enzymatic potential of actinobacteria from marine environments. Therefore, studies that gain insights into these understandings would be of great significance.

Among the wide range of enzymes, keratinases are of great interest in waste management. A large amount of keratin-derived wastes, such as feathers of poultry plants, are generated annually, leading to a severe environmental pollution problem. It is estimated that over 8.5 billion tons of feathers are produced worldwide [12]. Keratin represents nearly 90% of the total feather’s weight, which constitutes a great potential as a source of amino acids for biotechnological applications, including animal feed [13]. Currently, feathers are converted to feather meal through steam pressure and chemical treatment, which require high-energy input and can destroy essential amino acids [10,12]. A promising alternative to recover feather waste is the use of keratinases [14]. Their ability to degrade the recalcitrant protein keratin constitutes a remarkable property [15]. Microbial keratinases can be obtained from both fungi and bacteria [16], where among the latest, the Gram-positive group, such as *Bacilli* and *Actinobacteria*, have been recognized as important keratinase-producers. Among actinobacteria, several genera, such as *Actinomadura* [10], *Kocuria* [17], *Kytococcus* [18], *Microbacterium* [19], *Nocardiopsis* [20], *Streptomyces* [21,22,23,24], and *Thermoactinomyces* [25,26], have been described to have the capacity to produce keratinases.

The so-called keratinases have been defined as a subset of proteases, which have the capability of degrading insoluble keratin substrates, such as feathers, wool, nails, and hair [27]. These enzymes usually hydrolyze soluble proteins, such as casein, more effectively than insoluble proteins, such as keratins [28]. Most of them belong to the serine protease family [22,24,29], and a few are classified as metalloproteases [19,30]. Although keratinase purification and characterization have been the main focus of the investigation, there is still no clear distinction between keratinases and other common proteases. Besides some exceptions, purified keratinolytic proteases are often ineffective to hydrolyze native keratin [28,31]. In this sense, the term keratinase has been questioned because this enzyme does not respond to its specific substrate. On the other hand, due to the high content of cysteine in the keratin structure, the reduction of disulfide bonds seems to be essential for efficient keratin degradation. This process should include two main events, sulfitolysis (the breakdown of disulfide bonds) and proteolytic attack by keratinases [32]. Production of intra- and/or extracellular disulfide reductases [33], the release of sulfite and thiosulfate [34], and also a cell-bound redox system [35,36] are reported to lead to sulfitolysis. The mechanism of microbial keratinolysis is not yet completely elucidated, but considering these perspectives, keratinases can be considered proteases with keratinolytic activity, which, together with other keratinolytic enzymes or biologic factors, contribute to the keratin degradation [31]. Comprehensive research that considers different approaches can be essential to understand this process.

In this study, we explored the biocatalytic potential of seventy-five strains from our Chilean marine actinobacterial culture collection, previously isolated from marine sediments, sponges, and sea urchins collected from the coast of Chile [37,38,39], focusing mainly on the search for keratinolytic enzymes. First, a qualitative screening was performed with different substrates to evaluate the production of extracellular enzymes, such as cellulases, amylases, lipases, proteases, and keratinases. From this screening, we selected keratinolytic strains for an in-depth characterization of their keratinolytic potential, using tests of enzyme activity and keratin degradation. Our results showed that two streptomycete strains—*Streptomyces* sp. G11C, isolated from Penas Gulf, and *Streptomyces* sp. CHA1, isolated from Chañaral de Aceituno cove—had the highest keratinolytic activity and produced the release of thiol groups in the broth culture with feathers as the only carbon source. Subsequent characterization of the culture supernatants of both strains evidenced that they had similar behavior in the production of keratinolytic enzymes. Finally, to gain insights regarding which proteases are involved in keratin degradation, gel slices from protease zymograms of secreted proteins of *Streptomyces* sp. G11C, during growth with feathers as a keratin source, were identified by liquid chromatography-tandem mass spectrometry (LC-MS/MS). Together, these results provide new insight into the biocatalytic potential of marine-derived actinobacterial strains and contribute to elucidating some key factors underlying the mechanism of keratin degradation.

## 2. Results

### 2.1. Hydrolytic Enzyme Screening

A total of 75 marine-derived actinobacterial strains belonging to 29 genera, previously isolated from the Chilean coast [37,38,39], were screened for their ability to secrete hydrolytic enzymes on agar plates with the substrate corresponding to each enzymatic activity evaluated: tributyrin for lipase activity, skim milk for protease activity, gelatin for gelatinase activity, carboxymethylcellulose (CMC) for cellulase activity, starch for amylase activity, and turkey feather for keratinase activity. Additionally, the hemolytic activity was tested on blood agar plates (Figure 1). Enzyme activities were monitored by the diffused digested zone displaying hydrolysis, by diameter comparison of the digested zone versus the colony formed. In addition, the keratinolytic assay was complemented with a test in culture tubes with a feather as the sole source of carbon and nitrogen due to the difficulty of observing hydrolysis halos.

In total, 96% of the actinobacterial strains used in this study produced at least one hydrolytic enzyme under the conditions described. Overall, strains producing amylase and cellulase activities were the most abundant and diverse functional group, reaching 86.7% (65 strains belonging to 19 genera) and 84.0% (63 strains belonging to 17 genera) of the strains evaluated, respectively. Whereas, the strains producing hemolytic activity represented the smallest group, covering 34.7% (26 strains belonging to six genera) of the total strains assessed. In contrast, only a few strains showed no activity with the enzymatic tests evaluated in our study—a *Corynebacterium* and a potentially-novel strain of the *Nocardiopsaceae* family from Valparaíso and a *Nesterenkonia* strain from Rapa Nui. Interestingly, only one strain, *Salinibacterium* sp. VpJ6, showed one single strong activity corresponding to lipase, whereas other activities were negative under the conditions tested.

To compare the hydrolytic activity profile of actinobacterial strains, the enzymatic data, expressed as levels of enzymatic activity (LEA, halo ratio related to the colony), were analyzed to detect strains with strong enzymatic activity. The dataset was then normalized as a quaternary code, where “3” (depicted as a blue square) shows high activity, “2” (depicted as a green square) shows medium activity, “1” shows low activity (depicted as a light blue square), and “0” (depicted as a silver square) shows the absence of activity in a strain (Appendix A). To assess the relative distance of each enzymatic activity profile, hierarchical clustering was performed (Figure 2). The hierarchical cluster analysis showed that the strains could be roughly grouped into three distinct clusters according to their enzymatic profiles. One of these groups (Clade I) was comprised of 20 streptomycete strains, which had the strongest activities with at least four of the hydrolytic enzyme tests performed: gelatinase, keratinase, cellulase, and amylase. Interestingly, this clade was the group with the strongest keratinolytic activity and was the only clade with strains belonging to a single genus, which is *Streptomyces*. Only one strain (CHD11) branched separately compared to the rest of the strains in this specific clade.

Most of the 31 strains comprising the biggest clade (Clade II) in Figure 2 had a high cellulase activity and belonged to 10 genera: *Arthrobacter*, *Brachybacterium*, *Flaviflexus*, *Kocuria*, *Nesterenkonia*, *Streptomyces*, with stronger cellulase activity and *Knoellia*, *Micrococcus*, *Nocardiopsis*, *Tessarococcus*, *Serinococcus* with medium or low cellulase activity. One subgroup depicted in this cellulase clade, where most strains were *Streptomyces*, also had a strong activity with the rest of the enzyme activities tested; whereas the following subgroup comprised of *Kocuria*, *Micrococcus*, *Nesterenkonia*, and *Streptomyces* strains had a generally poorer activity with the rest of the evaluated enzymes. The third subgroup with strains belonging to *Brachybacterium, Flaviflexus, Knoellia, Kocuria*, and *Tessaracoccus* had even less activities with the rest of the enzymes, except with the amylase activity. The third clade (Clade III) from the hierarchical cluster analysis in Figure 2 was the group with less hydrolytic activity in our screening tests. This clade encompassed 24 actinobacteria belonging to 22 genera, with only nine strains with strong hydrolytic activity, including five strains with high lipase activity, one strain with high protease activity, and five strains with high gelatinase activity.

Our final aim was to prioritize strains considering their keratinolytic activity according to feather degradation in liquid culture for further studies. In this test, 30 strains belonging to the genera *Streptomyces*, *Micrococcus*, and *Nocardioides* showed some level of feather degradation. All these strains that had low, medium, and strong keratinolytic activity (Figure 2) were considered for the following analysis.

### 2.2. Screening of Keratinolytic Actinobacteria in Liquid Medium

A subset of 30 actinobacterial strains was used for further assessment in order to prioritize the selection of strains with the highest keratinolytic activity. Cultures were prepared in 250 mL flasks containing 60 mL of saline medium with turkey feathers as sole carbon and nitrogen source and subsequently grown for 5 days. Daily samples were taken to measure the keratinolytic and proteolytic activity, as well as the sulfhydryl group concentration. On the last day, culture volumes were filtered, and the insoluble material was dried to calculate the feather degradation percentage.

In this study, the selection of the strains with stronger keratinolytic potential was made, prioritizing the measurements of the feather degradation percentage—the keratinolytic activity, the presence of sulfhydryl groups, and finally the proteolytic activity (Appendix A). Maximum values obtained on day 5 of the culture are depicted in Figure 3. Twelve streptomycete strains evidenced a degradation percentage of more than 70%, presenting a high potential for the hydrolysis of keratin in feathers. These strains were isolated from different areas of the Chilean coast—*Streptomyces* sp. CHA16, CHA1, CHB9.2, CHB19.2 were isolated from Chañaral de Aceituno Island, *Streptomyces* sp. Vc17.3-30, Vc17.4, Vc714c-19, and Vc67-4 from Valparaíso bay, *Streptomyces* sp. IpFD-1.1 from Rapa Nui Island, *Streptomyces* sp. G11C from Penas Gulf, and *Streptomyces* sp. EL5 and EL9 from a sea urchin from Chiloé Island. However, some strains did not present keratinolytic activity or a significant sulfhydryl concentration in the medium. Particularly, one strain that was not selected, *Streptomyces* sp. Vc714c-19, presented both high keratinolytic and proteolytic activity. However, the aim of this study was to characterize enzyme(s) with specific keratinolytic activity. For this reason, strains that secreted proteases with little specificity to keratin were discarded. Finally, after a systematic analysis, the strains selected with the strongest keratinolytic activity were *Streptomyces* sp. CHA1, isolated from marine sediments obtained at 31.5 m depth from Chañaral de Aceituno Island in the northern area of Chile, and *Streptomyces* sp. G11C, isolated from marine sediments at 70 m depth, derived from Penas Gulf from the southern region of Chile [39].

Figure 4 shows graphs depicting the keratinolytic activity together with the accumulation of sulfhydryl groups during the five days of cultivation of *Streptomyces* sp. CHA1 and G11C using turkey feathers as the sole carbon and nitrogen source. Both streptomycete strains presented an increase in both measured variables, reaching a maximum of keratinolytic activity after five days of incubation: 61.3 U/mL for strain CHA1 and 60.0 U/mL for strain G11C (Figure 4A). In addition, sulfhydryl concentrations reached 245.7 µM for strain CHA1 and 284.5 µM for strain G11C (Figure 4B). The percentage of feather degradation after five days of incubation was 84% and 78%, respectively. This was consistent with Figure 4C, where both strains G11C and CHA1 exhibited complete feather degradation in liquid culture after seven days of incubation, whereas strain CHD11 presented low keratinolytic activity (41.7% of feather degradation) and partial dissolution of the feather in the liquid medium.

It is worth mentioning that strain CHD11 showed high feather degradation in tubes after 15 days of culture (qualitative tests, Figure 2), explaining the fact of the slight increase in the curves of keratinolytic activity and concentration of sulfhydryl groups up to 5 days of culture, evidencing a slower metabolism in this strain. These results were consistent with the *in-silico* analysis reported by Valencia et al. [40], who reported some putative proteases phylogenetically close to known keratinases in the genome of *Streptomyces* sp. CHD11.

### 2.3. Production of Keratinolytic Enzymes by Streptomyces sp. CHA1 and Streptomyces sp. G11C: Effect of Feather Concentration, Temperature, and Inoculum Percentage

For both selected strains—*Streptomyces* sp. CHA1 and *Streptomyces* sp. G11C—the effect of the following operational parameters was studied: feather concentration (5, 10, 15, and 20 g/L), incubation temperature (15, 20, 25, 30, and 37 °C), and inoculum percentage (2.3, 5, 7.5, and 10% *v*/*v*). These parameters were systematically investigated, varying one parameter at a time (Appendix A). 

The substrate concentration, incubation temperature, and inoculum size were found to be important parameters that influenced the production of keratinolytic enzymes. The optimum range of feather concentration for keratinolytic activity production was between 10 and 15 g/L for both *Streptomyces* strains CHA1 and G11C; although when 10 g/L of the feather was used, a decrease in the sulfhydryl group concentration was evidenced (Figure 5A,B). At higher and lower substrate concentrations, a decrease in keratinolytic activity, proteolytic activity, and sulfhydryl groups was observed. The feather degradation parameter did not present meaningful changes between 10 and 20 g/L of feathers, although in the culture of the strain CHA1 with a substrate concentration of 20 g/L, a higher percentage of degradation was observed. On the other hand, the keratinolytic activity and sulfhydryl groups concentration of both strains were maximal at an incubation temperature of 30 °C (Figure 5C,D). The degradation of feathers did not have significant changes between 20 and 37 °C. Interestingly, the proteolytic activity in the culture of both *Streptomyces* strains CHA1 and G11C was significantly higher to 20 °C and 25 °C, respectively, evidencing a greater proteolytic activity under these conditions, different from the production profile of keratinolytic enzymes. Besides, the increasing inoculum size (from 2.5 to 10% *v*/*v*) showed improved keratinolytic activity, with a significant effect at 10% v/v inoculum in both strains (Figure 5E,F). The feather degradation and sulfhydryl groups formed did not vary with the *Streptomyces* sp. G11C culture, but an increase in the proteolytic activity at 7.5% *v*/*v* inoculum was observed. With respect to *Streptomyces* sp. CHA1 culture, the sulfhydryl group concentration was higher at 7.5% *v*/*v* inoculum. Feather degradation and proteolytic activity did not show significant changes in this culture. In summary, the maximum keratinolytic activity production for *Streptomyces* sp. CHA1 was 63.5 U/mL, with a 78.4% of feather degradation on the fifth day of incubation at 30 °C, with 10% v/v inoculum and 15 g/L of feathers, and the maximum keratinolytic activity production for the *Streptomyces* sp. G11C was 78.3 U/mL, with 81.7% of feather degradation in the same conditions. The maximum values for each parameter are presented in Table 1.

### 2.4. Partial Characterization of Cell-Free Culture Supernatant Containing Crude Enzymes 

The partial characterization of culture supernatants was accomplished for both streptomycete strains: CHA1 and G11C. The temperature and pH profiles were evaluated using culture supernatants after five days of incubation with turkey feathers as the sole carbon and energy source (Appendix A). Keratinolytic and proteolytic activities were measured using keratin azure and azocasein as substrates, respectively. Temperature variations were performed between 20 and 90 °C, at pH 8, whereas the pH variations were between 3 and 11, at 37 °C.

Both strains presented a similar enzymatic profile, varying the temperature and pH of the reaction (Figure 6). Their enzymes were more active in the neutral to alkaline pH range, presenting a maximum enzymatic activity at pH 9, where *Streptomyces* sp. CHA1 presented 63.3 U/mL of keratinolytic activity and 12.4 U/mL of proteolytic activity (Figure 6A). *Streptomyces* sp. G11C presented 80.2 U/mL and 20.4 U/mL of keratinolytic and proteolytic activity, respectively (Figure 6B). On the other hand, the enzymes of both strains were active in a wide range of temperatures, between 20 and 90 °C, although a different behavior was evidenced for each substrate. Both strains presented a maximum of keratinolytic activity at 50 °C—106.1 U/mL for *Streptomyces* sp. CHA1 (Figure 6C) and 138.4 U/mL for *Streptomyces* sp. G11C (Figure 6D). Concerning the proteolytic activity, both strains demonstrated maximum activity at 70 °C—60.9 U/mL for strain CHA1 and 56.7 U/mL for strain G11C. The difference in the profiles of enzymatic activity versus temperature could be explained due to the presence of different enzymes (with different physicochemical properties) in the supernatants of both strains, which might be more specific for one substrate or another (casein or keratin azure). Under these conditions, both strains secreted a diverse set of proteases, as determined by a gelatin zymogram (Figure 6E). To deepen the understanding of the mechanism of keratin degradation, the bands from SDS-PAGE of *Streptomyces* sp. G11C, corresponding to the areas with proteolytic activity (clear zones), were subjected to a proteomic analysis to identify which proteases are expressed in the secretome.

### 2.5. Secretome Analysis of Streptomyces sp. G11C

To gain insights regarding the enzymes involved in the keratin degradation, gel slices corresponding to proteolytic bands of the secreted proteins of *Streptomyces* sp. G11C during growth with feathers, such as a keratin source, were identified by mass spectrometry (LC-MS/MS). Given that both *Streptomyces* strains CHA1 and G11C presented similar characteristics under the conditions analyzed, and according to their previous 16S rRNA gene identification [39], they were phylogenetically close. Therefore, *Streptomyces* sp. G11C, which evidenced the highest keratinolytic activity, was chosen to deepen our studies.

The secretome of the keratinolytic strain G11C presented the induction of 178 proteins identified by proteomics belonging to different protease families (Appendix A). These proteins included the following proteases of the MEROPS families: 67 serine (S) peptidases, 48 metallo (M) peptidases, nine cysteine (C) peptidases, two threonine (T) peptidases, one aspartic (A) peptidase, and two mixed (P) peptidases. The M6 family metalloprotease domain-containing protein G11C_02936 seemed to be the most abundant protein, registering the highest number of Spectral Counts (SC): 91. A recent bioinformatics analysis, based on a genomic comparison of *Streptomyces* sp. G11C, with two *Streptomyces* strains with differential keratinolytic activity, allowed the identification of a set of putative proteases that could potentially be involved in its keratinolytic capacity: seven putative keratinases and 17 putative proteases, unique to strain G11C [40]. In this study, we identified 18 of these proteases in the secretome, including six of the predicted putative keratinases (Table 2). These potential keratinases were mostly of the serine peptidase type, being the most abundant the S8 family peptidase G11C_03013 and the serine protease G11C_00267, with 79 and 60 SC, respectively. Besides, proteases with lower relative abundance (SC ≤ 20) were identified, and the corresponding genes were located in the same genetic context as genes encoding for other proteases of the secretome. This was the case for X-Pro dipeptidyl peptidase G11C_03889, the S8_S53 family peptidase G11C_04177, and the hypothetical protein G11C_03649 (Appendix A). According to our manual curation, this last sequence could not be classified, being finally annotated as a hypothetical protein. On the other hand, another group of potential keratinases, which presented a lower relative abundance (SC < 10), should be highlighted, presenting genetic contexts associated with different mechanisms of microbial degradation of keratin. This was the case of the calpain family cysteine protease G11C_03951, the PepSY domain-containing protein G11C_03746, and the S1 family peptidase G11C_05333, whose genetic contexts possessed genes for oxidoreductases enzymes (Appendix A). Different types of disulfide reductases or similar reducing forces had been identified in the microbial keratin hydrolysate, proposing their participation in the reduction of disulfide bridges present in the secondary structure of keratin [41,42]. It should be noted that the genetic context of S1 family peptidase G11C_05333 also had a gene that encodes a sulfite exporter TauE/SafE. Secretion of the reducing agent sulfite by dermatophytes keratinolytic microorganisms, which breaks keratin disulfide bonds, had also been reported [41,43,44,45]. In this sense, the genetic context of the extracellular serine proteinase G11C_02264 was also highlighted because it had genes related to sulfite metabolism (Appendix A). Finally, it should be mentioned that most of the proteases described in Table 2 can act as endopeptidases. Being most of them classified as serine endopeptidases, followed by metalloendopeptidases and cysteine endopeptidases. In less abundance, the presence of exoproteases, such as aminopeptidases and cysteine exoproteases, was also detected.

## 3. Discussion

Most of the marine-derived actinobacterial strains used in this study (96%) produced at least one hydrolytic enzyme under the conditions described, showing an enormous enzymatic diversity and biotechnological potential. The most abundant extracellular activities were both cellulase (86.7% of total strains, including 19 genera) and amylase (84.0% of total strains, including 17 genera). Carbohydrates, such as starch and cellulose, play an important role in biogeochemical cycles emerging in marine water columns and sediment-water interfaces [46,47]. Cellulolytic activity of marine actinomycetes is commonly observed [48,49,50,51]. In our study, strains that presented stronger cellulolytic activity belonged to the genera *Streptomyces*, *Kocuria*, *Arthrobacter*, *Nesterenkonia*, *Faviflexus*, and *Brachybacterium*, which was in agreement with other studies [52,53,54,55], with the exception of *Flaviflexus*. To our knowledge, this is the first report showing cellulolytic activity for a *Flaviflexus* strain. Strains that presented stronger amylase activity belonged to the genera *Arthrobacter*, *Faviflexus*, *Kocuria*, *Nocardiopsis*, *Serinicoccus*, *Streptomyces*, and *Tessaracoccus*. There are reports of actinobacterial genera that produce amylases, such as *Streptomyces* [56], *Micrococcus* [57], *Arthrobacter* [58], *Nocardiopsis* [59], and *Kocuria* [60]. Interestingly, there are no reports of characterized amylases from strains belonging to *Faviflexus* and *Tessaracoccus* genera. 

Enzyme activity profiles of our marine *Streptomyces* exhibited broad enzymatic diversity, showing stronger activities with at least four of the hydrolytic enzyme tests performed: gelatinase, keratinase, cellulase, and amylase. These strains had the exceptional ability to degrade complex molecules. Interestingly, half of the streptomycete strains grouped in clade I (Figure 2) were derived from Chañaral de Aceituno Island, where three of the four strains belonging to Rapa Nui and the only strain evaluated from Penas Gulf were also present. One characteristic that shared all three Chilean regions was the presence of baleen whales [61,62,63]. Baleen whales have baleen that are keratinous plates attached to the upper jaw, similar to a comb rack, permitting the filtering of seawater for planktonic aggregations [64]. These structures are largely made up of α-keratin and usually erode due to mechanical and hydrodynamic wearing [65]. All this supports the idea that keratinous waste may be present in marine sediments derived from Chañaral de Aceituno, Rapa Nui, and Penas gulf, and therefore the presence of keratinolytic bacteria seems feasible. Few studies have been conducted to search for keratinolytic bacteria from marine environments. Herzog et al. [66] isolated keratin-degrading bacteria, mostly belonging to the *Bacillus* genera, which is a well-known feather-degrading bacterium [67]. Our work evidenced that marine habitats are a promising source of keratinolytic actinobacteria.

Strains present in clade II (Figure 2) mostly arose for manifesting high cellulase activity, followed by high amylase and gelatinase activity. In general, the presence of cellulases and amylases was well distributed among different genera, whereas gelatinase activity was less common, according to our study. It should be said that, in general, the gelatin agar was better than the skim milk agar for the protease test, as evidenced in other works [68]. As for clade III, it presented a group of strains (24 strains belonging to 22 genera) with less hydrolytic activities. Two subgroups, within this clade, with high gelatinase and lipolytic activity, emerged. Stronger gelatinase activity was detected in *Janibacter*, *Kytococcus*, *Nocardiodes*, *Curtobacterium,* and *Streptomyces* strains (along with *Arthrobacter* and *Kocuria* strains from clade II). Proteolytic strains belonging to these genera have been previously reported [18,69,70,71,72,73]. Finally, strains of clade III with stronger lipolytic activity belonged to the *Brevibacterium*, *Janibacter*, *Microbacterium*, *Pseudonocardia,* and *Salinibacterium* genera. Up to now, lipolytic activities from various terrestrial actinobacteria belonging to *Brevibacterium*, *Microbacterium,* and *Streptomyces* strains [74,75,76] have been reported; however, lipolytic activities derived from marine microorganisms are limited [77,78]. According to our knowledge, there are no reports of characterized lipases from actinobacterial strains belonging to *Pseudonocardia*, *Salinibacterium*, and *Serinicoccus* genera. Taken together, our results suggested that rare actinobacteria are a promising source for the discovery of novel enzymes of industrial interest.

A subset of 30 marine-derived actinobacterial strains evidenced keratinolytic activity on agar plates and liquid culture with feathers. These strains belonged to three genera previously reported to degrade keratin: *Nocardiodes*, *Micrococcus*, and *Streptomyces* [22,79,80]. Overall, the keratinolytic activity profile of all strains evaluated varied in comparison to the proteolytic activity profile, suggesting the presence of different proteases, specifically for each substrate. The so-called keratinases are proteases with specificity towards recalcitrant keratin substrates, different from conventional proteases that can hydrolyze soluble and more accessible substrates, such as casein [32]. For example, *Micrococcus* sp. IpBE2 showed high proteolytic activity but presented a low percentage of feather degradation and low keratinolytic activity, suggesting that this strain is a good source of conventional proteases but not of keratin-degrading proteases. Surprisingly, for some actinobacterial strains, feather degradation was not directly associated with the measured keratinolytic activity. This was the case for *Streptomyces* strains CHC8 and CHC16 isolated from Chañaral de Aceituno Island and *Streptomyces* strains H-KF8 and H-CB3 from Comau fjord. This is possible because the keratinolytic activity assays were performed using azure-tagged sheep’s wool (Keratin azure) as a substrate, which mainly includes α-keratin [81], unlike the feathers that contain a higher proportion of β-keratin [82]. Due to the more extended conformation of β-sheets compared to α-helices, the former is usually more accessible to the action of keratinases than the latter [28]. In this case, the strains mentioned above could have greater specificity towards feather keratin compared to wool keratin. In addition, all the strains showed the presence of sulfhydryl groups within the culture broth, suggesting the presence of other factors or enzymes that are acting in the keratin degradation.

*Streptomyces* sp. CHA1 and *Streptomyces* sp. G11C presented a significant feather degradation percentage and higher keratinolytic activities, demonstrating a high potential for the recovery of feather waste. Substrate concentration, incubation temperature, and, to a lesser extent, inoculum size were found to be important parameters that influenced the keratinolytic activity in both strains CHA1 and G11C. At higher and lower substrate concentrations, with respect to 15 g/L of feathers, a decrease in keratinolytic activity, proteolytic activity, and sulfhydryl groups was observed. It has been previously shown that elevated protein concentrations may cause substrate inhibition of proteolytic enzymes [26,83]. Besides, inhibition due to product formation cannot be discarded. Product inhibition becomes more relevant when incubation times are longer, as in our case [83]. On the other hand, both strains showed maximum keratinolytic activity at 50 °C and pH 9. The optimal temperature for keratinolytic enzymes may be very variable, often depending on the source of origin. Generally, these enzymes show an optimal activity between 40 and 70 °C [28]. The alkaline protease of the marine *Streptomyces fungicidicus* MML1614, isolated from the Bay of Bengal, India, is optimally active at 40 °C and pH 9 [7]. Maximum activity at 65 °C is also recorded using the crude enzyme supernatant for some marine *Bacillus* strains [66]. With respect to the keratinolytic activity profile versus pH, most of the microbial keratinolytic enzymes are alkaline or neutral proteases with a pH optimum ranging 7.5–9.0 [28], which are in agreement with our results.

Finally, to obtain insight into the mechanism of keratin degradation mediated by streptomycetes, we identified the proteins secreted by *Streptomyces* sp. G11C in a culture with feathers as a source of keratin. *Streptomyces* species have been recognized as one of the main producers of keratinases [28,41]. Most studies on keratinolytic actinomycetes have focused on the characterization of isolated keratinases [14,21,22,23,24,84,85,86]. However, purified keratinases known to date cannot completely solubilize native keratin, which suggests that a pool of keratinolyitic proteases is needed to penetrate and break down keratin structure instead of a single enzyme [27,31]. Pathogenic fungi are known to secrete a wide variety of proteases to decompose keratin-rich substrates, mainly endoproteases from families A1, S8A, M36, and M35 [87]. An investigation of the non-pathogenic fungus *Onygena corvina* has suggested that a combination of endoprotease (S8), exoproteases (M28), and oligopeptidase/metalloprotease (M3) may act synergistically to break down pig bristle keratin [88]. In addition, four additional components have been identified to act in synergy with these proteases: lytic polysaccharide monooxygenases (AA11/LPMOs), disulfide reductases, cysteine dioxygenases, and sulfite [31]. Based on the complex structure of keratin and its high cysteine content, it has been suggested that its microbial degradation should include two main stages: sulfitolysis or breakage of disulfide bonds and proteolytic attack by keratinases [32]. Breakage of disulfide bonds might occur due to the action of disulfide reductases, the release of sulfite/thiosulfate, and cell-bound redox systems [82]. However, this reduction is poorly understood for most keratinolytic microorganisms so far. In some bacteria, the synergistic action of various enzymes in keratin degradation has been reported. In *Stenotrophomonas* sp. strain D-1, the cooperative action of two enzymes—a serine protease and a disulfide bond-reducing protein—leads to effective degradation of keratin [33]. *Bacillus subtilis* CH-1 presents four kinds of enzymes involved in keratin degradation, including extracellular protease Vpr, peptidase T, γ-glutamyl transpeptidase, and glyoxalmethylglyoxal reductase [89]. Huang et al. [90] reported the presence of five proteases in the culture of *Bacillus* sp. 8A6 with keratin-rich substrates, belonging to four protease families M12, S01A, S8A, and T3. Recently, Li et al. [91] reported through a transcriptome analysis that *Streptomyces* sp. SCUT.3 overexpressed a set of protease-coding genes from various families, together with genes encoding for oxidoreductases and genes related to sulfite metabolism.

Recently, Valencia et al. [40] predicted keratinolytic proteases in the genome of *Streptomyces* sp. G11C, based on genomic comparison with other *Streptomyces* strains with differential keratinolytic activity. In our study, 18 of these in silico predicted proteins were identified in the secretome of this strain (Table 2). Analysis of the genetic contexts of some of these proteins determined that their corresponding genes were close to those encoding for other proteases, also identified in the secretome (Appendix A). The presence of keratinolytic proteases and other types of proteases in the secretome of *Streptomyces* sp. G11C grown in the presence of turkey feather suggested a synergistic proteolytic action for the degradation of keratin. On the other hand, the presence of genes for oxidoreductases and for the metabolism and export of sulfite within the genetic contexts of these secreted keratinases suggested that the degradation of keratin carried out by keratinases and proteases could be complemented by the secretion of oxidoreductases and sulfite, which have high reducing power for disulfide bridges. To propose a possible mechanism of keratin degradation in *Streptomyces* sp. G11C, potential keratinases and proteases were selected that, according to Valencia et al. [40], presented extracellular properties according to bioinformatics prediction. In addition, the protease G11C_02936 was included, which, although it was not bioinformatically predicted as a potential keratinase, is interesting because it is the most abundant in the secretome and because it belongs to the M6 family, which is a family of proteases not reported so far in keratinolytic studies. A model of the extracellular degradation of keratin in this strain is observed in Figure 7. This included the potential extracellular keratinases G11C_03013, G11C_02264, and G11C_05333, the proteases G11C_02936 and G11C_05332, the oxidoreductase G11C_02263, the sulfite metabolism proteins G11C_02252 to G11C_02261, and the sulfite exporter G11C_05337 (Figure 7A). It should be noted that the potential keratinase G11C_05333 was in the same genetic context as the membrane protease G11C_05332 and the sulfite exporter G11C_05337, and that the potential keratinase G11C_02264 was in the same genetic context as the oxidoreductase G11C_02263 and sulfite metabolism enzymes G11C_02252-02261 (Figure 7B).

In our study, the biocatalytic potential of 75 marine-derived actinomycetes, isolated from the Chilean coast, was addressed. These strains, including rare actinobacteria, were screened for extracellular enzyme activity, showing great potential for the production of cellulases, amylases, lipases, proteases, and keratinases. Our work confirmed, for the first time, the presence of amylases in *Tessarococcus* and *Flaviflexus* strains, the latter also with cellulolytic activity, whereas a strong lipolytic activity was evidenced by *Pseudonocardia*, *Salinibacterium*, and *Serinicoccus* strains. These rare actinobacteria can be novel sources of industrially relevant enzymes. Based on the analysis of 30 actinobacterial strains that exhibited significant keratinolytic activity, two streptomycete strains with stronger keratinolytic activity were selected for further characterization—*Streptomyces* sp. CHA1 and *Streptomyces* sp. G11C—evidencing the presence of alkaline keratinolytic enzymes. Finally, we identified a set of keratinolytic proteases secreted by *Streptomyces* sp. G11C, belonging mainly to the serine and metallo-superfamilies, the most abundant proteases being a serine protease from family S08 and a metalloprotease from family M06. This is the first study to report the participation of a bacterial protease of the family M06 in the keratin degradation. Furthermore, the *in-silico* study of the genetic contexts of some of these secreted proteases suggested the possible participation of oxidoreductases and sulfite that could be helping to break the disulfide bonds of keratin. Taken together, keratin degradation seemed to be possible with a set of multiple enzymes, and our results contributed to elucidate the possible mechanism of keratinolysis mediated by *Streptomyces* sp. G11C.

## 4. Materials and Methods

### 4.1. Actinobacterial Strains

A total of 75 marine actinobacterial strains belonging to 29 genera were screened for their ability to produce extracellular enzyme activities. These strains were previously isolated from various regions of the Chilean coast: Chañaral de Aceituno Cove, Rapa Nui, Valparaíso bay, Comau fjord, and Penas Gulf [37,38,39]. A selection was based on the genera described in the literature to have keratinolytic activity, including strains belonging to the following genera: *Actinomadura*, *Arthrobacter*, *Microbacterium*, *Nesterenkonia*, *Nocardiopsis*, *Streptomyces*, and *Kocuria*. Besides, one representative strain belonging to other genera from our culture collection was additionally tested. These were *Aeromicrobium*, *Agrococcus*, *Brachybacterium*, *Brevibacterium*, *Corynebacterium*, *Curtobacterium*, *Dietzia*, *Flaviflexus*, *Gordonia*, *Isoptericola*, *Janibacter*, *Knoellia*, *Kytococcus*, *Micrococcus*, *Nocardiodes*, *Ornithinimicrobium*, *Pseudonocardia*, *Rhodococcus*, *Salinactinospora*, *Salinibacterium*, *Serinicoccus*, and *Tessaracoccus*.

### 4.2. Primary Screening of Extracellular Hydrolytic Enzymes on Solid Media

For the detection of extracellular enzymatic activity, assays were performed on agar plates with the respective substrate, using the drop spot technique. Details of the culture medium used for each enzymatic activity are described below. All media were autoclaved at 121 °C for 20 min. To prepare the pre-inoculum, actinobacterial strains were incubated at 30 °C for a week on Tryptic Soy Agar (TSA) medium (Difco, NJ, USA). Subsequently, biomass was resuspended in 500 μL of a 0.85% NaCl solution, and 10 μL aliquot of each actinobacterial test strain was spotted onto appropriate media. In each Petri dish, four bacterial strains were evaluated. Agar plates for enzymatic activities were incubated at 20 °C for 7 days unless otherwise stated. The results were qualitatively expressed as levels of enzymatic activity (LEA) using the formula LEA = clearance zone diameter/colony diameter in millimeters. No activity is defined as LEA = 0, low activity is defined as 1 ≤ LEA < 2, medium activity is defined as 2 ≤ LEA < 3, and high activity is defined as LEA ≥ 3 [92]. 

#### 4.2.1. Amylase Activity

Amylase activity was determined with a starch agar-based medium containing 2 g of soluble starch, 5 g of peptone, 1 g of yeast extract, and 20 g of agar per liter. After incubation for 1 week, all Petri dishes were flooded with 3 mL of Lugol’s iodine solution (consisting of 1.0 g of iodine, 5.0 g of potassium iodide, and 330 mL of distilled water). A clear zone around the colony indicated hydrolysis of the starch present [92]. 

#### 4.2.2. Cellulase Activity

For cellulase activity, a carboxymethylcellulose (CMC)-based medium was prepared, containing 2 g of CMC, 2 g of NaNO_3_, 1 g of K_2_HPO_4_, 0.5 g of MgSO_4_, 0.5 g of KCl, 0.2 g of peptone, and 18 g of agar per liter. After incubation, the plates were flooded with 3 mL of Lugol’s iodine solution. The clear halo around the colonies indicated cellulase activity [93].

#### 4.2.3. Gelatinase Activity

Gelatinase activity was tested using agar plates containing 12 g of gelatin, 1 g of yeast extract, 4 g of peptone, and 18 g of agar per liter. Clear zones around the bacterial growth were considered as an indication of gelatinase activity [94].

#### 4.2.4. Lipase Activity

Lipase activity was assessed using the Spirit blue agar media (Difco, NJ, USA) composed of 10 g of casein enzymatic hydrolysate, 5 g of yeast extract, 17 g of agar, and 0.15 g of spirit blue dye per liter. A total of 35 g of spirit blue agar was used per liter of distilled water. Once the medium was at a temperature of about 50 °C after autoclave, tributyrin (3 mL/L) was added as a substrate and vigorously stirred before pouring into Petri dishes. Plates were inoculated and incubated at 20 °C for 3 days. Lipolysis was evidenced by halos observed around each bacterial colony, indicating that microorganisms metabolized the lipids.

#### 4.2.5. Protease Activity

Proteolytic activity was tested in skim milk plates containing 10 g of skim milk, 1 g of yeast extract, and 20 g of agar per liter. Clear zones around the growth were considered as an indication of proteolytic activity [92]. 

#### 4.2.6. Hemolytic Activity

The strains were tested for hemolytic activity on blood agar plates (Insumolab, Santiago, Chile). Clear zones around the growth after a week were considered as an indication of hemolytic activity.

#### 4.2.7. Keratinase Activity

Initially, keratinolytic activity was assessed using a medium described in one of our previous studies [40]. The medium consisted of 20 g of chopped turkey feathers, 2 g of NaCl, 0.5 g of MgSO_4_*7H_2_O, 0.27 g of KH_2_PO_4_, 0.35 g of K_2_HPO_4_, 18 g of agar, and 10 mL of a trace element solution containing 27 mM CaCl2, 4 mM Fe(III) citrate, 1.3 mM MnSO4, 0.7 mM ZnCl2, 0.16 mM CuSO4, 0.17 mM CoCl2, 0.10 mM Na_2_MoO_4_, and 0.26 mM Na_2_B_4_O_7_ per liter (pH 7.2). Turkey feathers used for the tests, obtained from a local poultry slaughterhouse, were washed several times and then ground on a Talsa PSV C15 cutter (Valencia, Spain) and an Omega TL32 mincer (Bologna, Italy) to reduce particle size. Chopped turkey feathers were dried at 60 °C for 24 h before use in the preparation of the culture medium. Feather agar plates with strains were incubated at 20 °C for a week. Due to the difficulty of visualizing the degradation halos, the actinobacterial strains that presented the main growth (diameter > 11 mm) were considered positive for keratinolytic activity screening. To determine if these strains effectively degraded keratin from feathers, they were inoculated in culture tubes with a liquid medium containing the same saline composition described above (without agar) and a feather as the sole source of carbon and nitrogen (Figure 1). The tubes were incubated for 15 days at 30 °C and 200 rpm. Strains that showed some degree of feather degradation were selected for further studies.

### 4.3. Growth Conditions for Keratinolytic Activity in Liquid Medium

A suitable volume of pre-inoculum of each actinobacterial strain in Tryptic Soy Broth (TSB) medium (Difco, NJ, USA) was added to a 250-mL Erlenmeyer flask with 60 mL of liquid medium containing trace salts with turkey feather as carbon and nitrogen source. The growth medium used was the same as mentioned above. For the comparison, the inoculum of each actinobacterial strain was standardized at an OD600 of 0.2. Cultivations were performed for 5 days at 30 °C and 200 rpm. A daily sample was taken for the measurement of keratinolytic activity, proteolytic activity, and determination of sulfhydryl groups. The samples were centrifuged (13,000 rpm for 10 min) to remove mycelia and medium debris, and the cell-free supernatant was stored at −20 °C until use. The culture supernatant containing crude enzymes was used in subsequent studies.

### 4.4. Quantification of Keratinolytic Activity

Keratinolytic activity was determined using a modified method of Letourneau et al. [95] using keratin azure (Sigma-Aldrich, St. Louis, MO, USA) as a substrate. The reaction mixture contained 0.2 mL of culture supernatant and 0.8 mL of 0.4% *w*/*v* keratin azure in 50 mM potassium phosphate buffer, pH 8. The mixture was shaken (350 rpm) for 1 h at 37 °C. After incubation, the mixture was kept on ice for 15 min, followed by centrifugation at 13,000 rpm for 10 min to remove the unutilized substrate. The supernatant was spectrophotometrically measured for the release of the azo dye at 595 nm. Control was kept with buffer and substrate without enzyme. One unit of keratinolytic activity was defined as the amount of enzyme causing an increase of 0.01 absorbance units in an hour under the given conditions.

### 4.5. Quantification of Proteolytic Activity

Proteolytic activity was measured by the modified method of Thys and Brandelli [96] using azocasein as substrate. The reaction mixture contained 0.1 mL of culture supernatant and 1 mL of 0.25% *w*/*v* azocasein in 50 mM potassium phosphate buffer, pH 8. The mixture was incubated at 37 °C for 5 min, and the reaction was stopped by adding 0.5 mL of 10% (*w*/*v*) trichloroacetic acid (TCA). The mixture was then centrifuged at 13,000 rpm for 10 min. Subsequently, 0.8 mL of the supernatant was added to 0.8 mL of 1N NaOH solution, and the absorbance at 430 nm was measured. One unit of proteolytic activity was defined as the amount of enzyme that resulted in an increase of absorbance at 430 nm of 0.1 units under the assay conditions used.

### 4.6. Determination of Sulfhydryl Groups 

Free sulfhydryl groups were spectrophotometrically determined at 412 nm by measuring the yellow-colored sulfide formed upon reduction of 5,5′-dithio-bis-(2-nitrobenzoic acid), also known as DTNB [97]. A DTNB solution produces a measurable yellow-colored product when it reacts with sulfhydryls. The reaction mixture contained 0.2 mL of culture supernatant, 1.7 mL of 20 mM potassium phosphate buffer (pH 6.6), and 0.1 mL of 10 mM DTNB in methanol. The absorption of the mixture was measured at 412 nm after 5 min of reaction. The concentration of sulfhydryl groups was estimated by comparison to a standard curve composed of known cysteine concentrations.

### 4.7. Percentage of Feather Degradation

The feather degradation was quantified in terms of the solubilization degree. After completion of the incubation period, the culture broth was filtered using pre-weighed Whatman grade 3 filter paper. The filter paper, along with insoluble material, was dried for 48 h at 60 °C until constant weight. The percentage of feather degradation was calculated based on the difference between the weight of the analysis culture and the final weight of the control culture (medium with feathers and without bacteria).

### 4.8. Protein Determination

Protein concentration was determined by the Bradford method [98] using the Bio-Rad assay reagent according to the manufacturer’s description with bovine serum albumin as the standard (Bio-Rad, Hercules, CA, USA).

### 4.9. Characterization of Cultural Conditions for Keratinolytic Activity

Production of keratinolytic enzymes and feather degradation by selected strains was further studied varying different culture conditions, using the methodology known as “one-variable-at-a-time”. The same saline medium described above was used, with an initial pH of 7.2. The effect of substrate amount was determined varying the feather concentration in the medium between 5 and 20 g/L (5, 10, 15, and 20 g/L), the effect of temperature with incubations between 15 and 37 °C (15, 20, 25, 30, and 37 °C), and the effect of inoculum amount was determined varying its volume between 2.5 and 10% *v*/*v* (2.5, 5, 7.5, and 10% *v*/*v*). Cultures were performed for five days at 200 rpm. Samples were taken during the fifth day of cultivation (where maximum activity was initially observed) to measure the keratinolytic activity, proteolytic activity, concentration of sulfhydryl groups, and the percentage of feather degradation.

### 4.10. Partial Characterization of Culture Supernatants: Influence of Temperature and pH

The effect of pH on keratinolytic and proteolytic activity was determined by incubating the reaction mixture at various pH ranges from 3 to 11 at a constant temperature of 37 °C. The following buffers were used at 100 mM: citrate buffer for pH 3–5, potassium phosphate buffer for pH 6–8, bicarbonate-carbonate buffer for pH 9–11. The effect of temperature on enzyme activity was measured by incubating the reaction mixture at temperatures from 30 to 90 °C (30, 40, 50, 60, 70, 80, 90 °C) at pH 8. The keratinolytic activity and proteolytic activity were measured using keratin azure and azocasein as substrate, respectively, according to the method described above.

### 4.11. Non-Reducing SDS-PAGE 

To analyze the secretome of the selected strains, proteins of the culture supernatants were concentrated 10-fold by an ultrafiltration process using an HP4750 stirred cell (Sterlitech, Kent, WA, USA) with a membrane of 10 kDa molecular weight cut-off. Later, proteins were separated by non-reducing sodium dodecyl sulfate-polyacrylamide gel electrophoresis (SDS-PAGE), according to the Laemmli method [99], using 12% polyacrylamide resolving gel. Protein samples were not treated with 2-mercaptoethanol or with high heat. Electrophoresis was carried out at 4 °C at constant voltage (80 V) for 3.5 h using a Bio-Rad Mini Protean 3 electrophoresis apparatus. Initially, samples were loaded on two identical gels, which after electrophoresis, received a separate treatment. One of them followed the conventional protein electrophoresis procedure: it was stained with 0.25% *w*/*v* Coomassie brilliant blue R-250 for 1 h and then decolorized with a destaining solution composed of methanol, acetic acid, and water (40:10:50) until bands were visualized. The other gel was reserved for zymography analysis. The molecular weight of the bands was estimated by comparing the relative mobility of proteins with known molecular weight markers ranging from 10 to 250 kDa (Bio-Rad).

### 4.12. Zymography

Activity staining of proteases present in the secretome was performed by zymography [100] using gelatin (20 mg/mL) as a substrate in 12% polyacrylamide resolving gel. After SDS-PAGE electrophoresis, the reserved gel was immersed in refolding buffer (2.5% Triton-X 100 in 50 mM Tris, pH 7.4) thrice for 20 min with gentle orbital shaking to remove SDS and thoroughly rinsed with double distilled water to remove Triton-X 100. Subsequently, proteins on the resolving gel were electrophoretically transferred to another polyacrylamide gel containing gelatin as a protein substrate [101]. Both resolving and receiving gels were assembled in a ‘‘sandwich’’ similar to the assembly for a Western blot. Then, electrophoretic transfer was carried out in a buffer containing 25 mM Tris and 192 mM glycine (pH 8.3) at 4 °C and constant voltage (20 V) for 20 min using a Bio-Rad Mini Trans-Blot cell. Enzyme activity was visualized by incubating the gel overnight in proteolysis buffer (50 mM Tris, 200 mM NaCl, 0.5 mM ZnCl2, 5 mM CaCl2 x 2H2O, and 3 mM NaN, pH 7.4) at 37 °C, followed by staining with 0.25% (*w*/*v*) Coomassie Brilliant Blue R-250 for 1 h and destaining until clear bands indicating proteolytic activity became visible.

### 4.13. Proteomic Analysis

Gel slices that showed proteolytic activity were excised and subjected to in-gel trypsin digestion, as previously described [102], and analyzed by liquid chromatography in-line to tandem mass spectrometry (LC-MS/MS). The resulting peptides were eluted in a 100 μm i.d. microcapillary column with a 5-μm pulled tip packed with 11 cm of 3-μm Aqua C18 resin (Phenomenex, Torrance, CA, USA) and directly analyzed in an LTQ-Orbitrap XL mass spectrometer (Thermo Finnigan, San Jose, CA, USA) using a 120-min single-phase step separation. The gradient profile consisting of 5 min in 100% (*v*/*v*) buffer A (5% acetonitrile/0.1% formic acid), 70 min in 15%, 30 min in 45%, 5 min in 75%, and 10 min in 100% (*v*/*v*) of a buffer B (80% acetonitrile/0.1% formic acid solution) was used. A cycle of one full-scan mass spectrum (300−2000 m/z), followed by five data-dependent MS/MS spectra at a 35% normalized collision energy, was repeated continuously throughout each step of the multidimensional separation. To prevent repetitive analysis, dynamic exclusion was enabled with a repeat count of 1, a repeat duration of 30 s, and an exclusion list size of 200. The application of mass spectrometer scan functions and HPLC solvent gradients was controlled by the Xcalibur data system (Thermo, Waltham, MA, USA). MS/MS spectra were extracted using the RawXtract program (version 1.9.9) and searched using PatternLab for Proteomics (www.patternlabforproteomics.org/) against sequences from the *Streptomyces* sp. G11C genome (WGS accession number JABTTT000000000) [40] with a 0.05% of protein False Discovery Rate (FDR). Appendix A shows the raw data of the proteins identified in each band analyzed.

### 4.14. Analysis In Silico of Genetic Contexts of Some Selected Peptidases

A select group of proteins was analyzed in silico to study their genetic contexts. The genes for these proteins and their adjacent genes were cured manually by searching for conserved domains and catalytic residues using the Conserved Domains Database (CDD) (http://www.ncbi.nlm.nih.gov/cdd). According to this, the names of some genes whose automatic annotation did not coincide with the analysis of conserved domains were rewritten.

### 4.15. Statistical Analysis 

All experiments were carried out as independent cultures in triplicate, and the mean values with standard deviation were presented. For comparisons between various culture conditions, data were analyzed by one-analysis of variance (ANOVA) with Turkey–Kramer multiple comparisons test, using the GraphPad Prism 7 software [103]. Differences were considered significant when *p* < 0.05.

## Figures and Tables

**Figure 1 marinedrugs-18-00537-f001:**
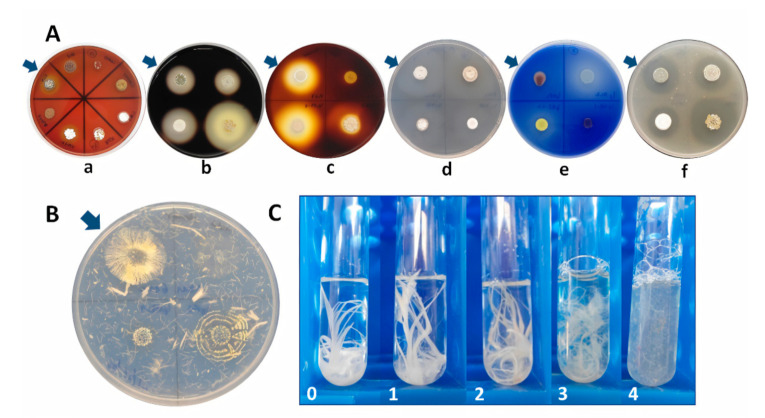
Colonies of representative actinobacterial strains grown on agar plates where enzymatic activities were assessed by the diffused digested zone displaying hydrolysis and growth in liquid culture with turkey feathers. (**A**) Representative actinobacterial strains grown on agar plates to determine (a) hemolytic activity: *Streptomyces* sp. EL9, EL5, CHD11, CHC141, H-CB3, H-KF8, VA42-3, and VH47-3; (b) amylase activity: *Streptomyces* sp. H-KF8, H-CB3, VA42-3, and VH47-3; (c) cellulase activity: *Streptomyces* sp. Vc67-4, VS4-2, Vc17.4, and Vc17.3-30; (d) gelatinase activity: *Streptomyces* sp. Vc714c-19, G11C, Vc74A-19, and Vc74B-19; (e) lipase activity: *Kocuria* sp. IpDF2, *Kytococcus* sp. IpAL2, *Kocuria* sp. IpEB1, and *Kocuria* sp. IpBJ-1.2; (f) protease activity (1% milk): *Streptomyces* sp. H-KF8, H-CB3, VA42-3, and VH47-3. Arrows indicate the first strain mentioned and then continuing clockwise. (**B**) keratinolytic activity: colonies from *Streptomyces* sp. strains G11C, IpFD-1.1, IpFD-6, and IpHD-1 grown on solid medium with feathers. (**C**) keratinolytic activity in liquid medium with streptomycete strains depicting the various degrees of feather degradation: (0) Negative control with medium containing one feather and no bacteria; (1) strain VS4-2 with no activity (intact feather); (2) strain Vc67-4 with low activity; (3) strain CHA3 with medium activity, and (4) strain CHA1 with high activity (completely degraded feather).

**Figure 2 marinedrugs-18-00537-f002:**
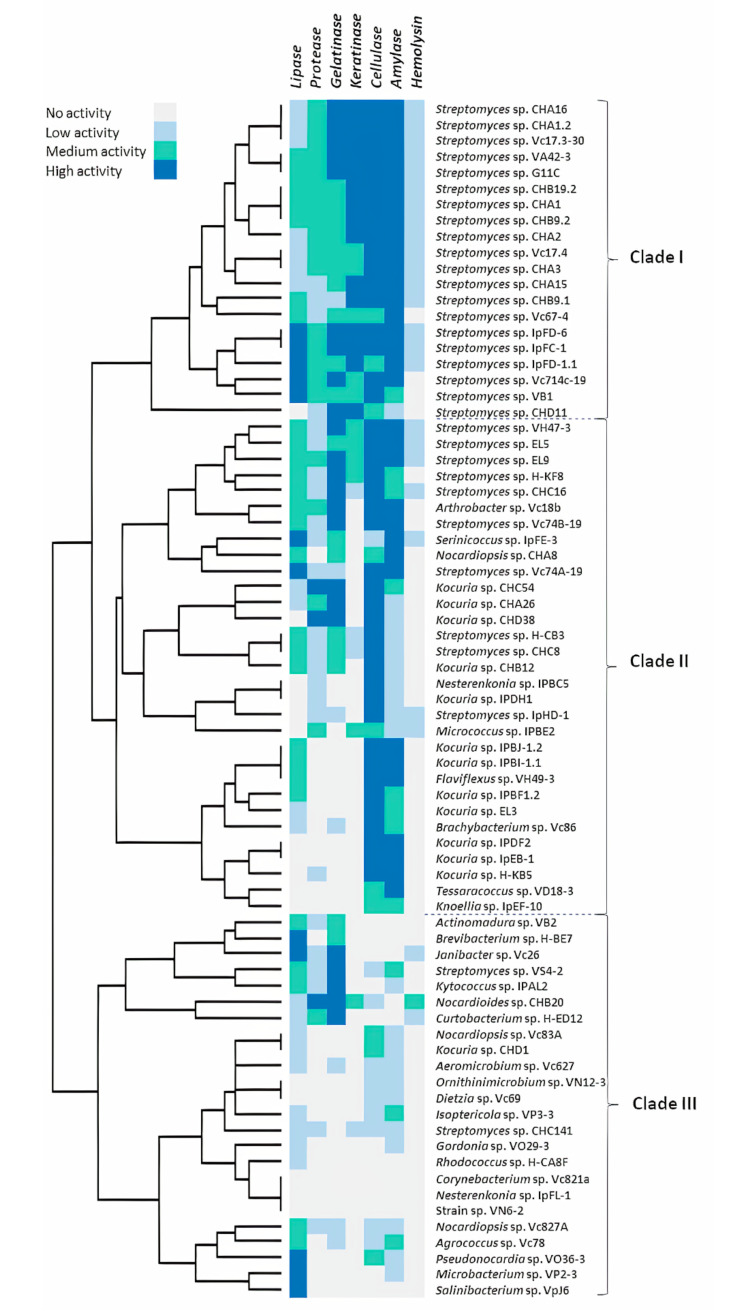
Bioprospecting for extracellular hydrolytic enzymes from marine actinobacteria. Heat map representation and a hierarchical cluster analysis based on the extracellular hydrolytic activity profiles. Cell color represents the activity level of the clustered actinobacterial strains: blue, high activity; green, medium activity; light blue, low activity; silver, no activity.

**Figure 3 marinedrugs-18-00537-f003:**
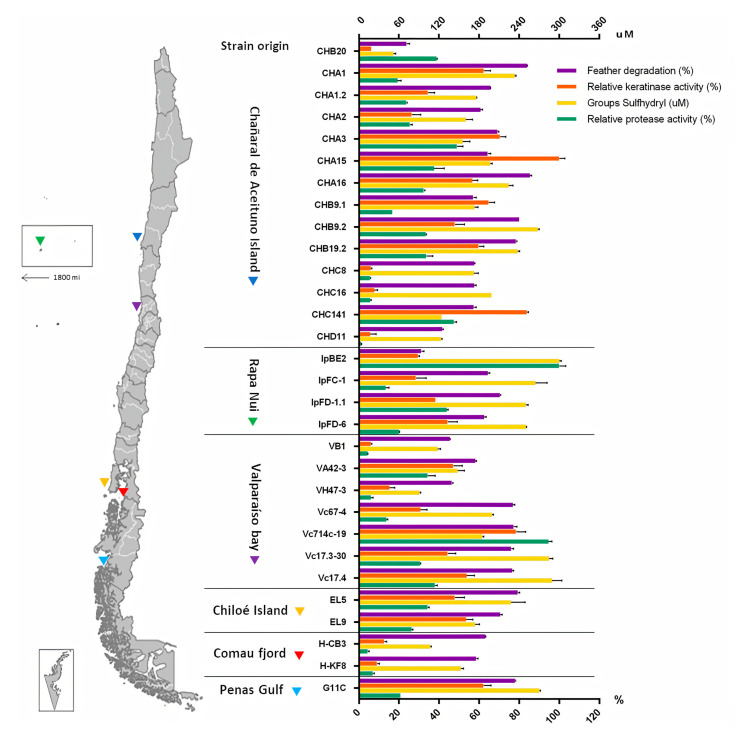
Origin of actinobacterial strains and their respective screening tests related to feather degradation (%), relative keratinolytic activity (%), sulfhydryl groups (µM), and relative proteolytic activity (%) in liquid culture containing turkey feathers (2% *w*/*v*) after 5 days of incubation. Most of the strains belonged to the *Streptomyces* genus, with the exception of strains IpBE2 and CHB20, which belonged to the *Micrococcus* and *Nocardioides* genera, respectively.

**Figure 4 marinedrugs-18-00537-f004:**
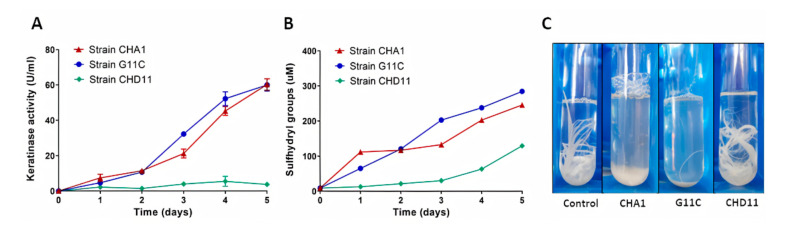
Keratinolytic activity, determination of sulfhydryl groups, and visual disintegration of turkey feathers during the culture of *Streptomyces* sp. CHA1, G11C, and CHD11. (**A**) Increase in the keratinolytic activity (U/mL) during the culture of *Streptomyces* sp. CHA1, *Streptomyces* sp. G11C, and *Streptomyces* sp. CHD11; (**B**) Presence of sulfhydryl groups (µM) during the culture of *Streptomyces* sp. CHA1, G11C, and CHD11; (**C**) Visual disintegration of turkey feathers during the culture of *Streptomyces* sp. CHA1, G11C, and CHD11 after 7 days of incubation [40]. Negative control with medium containing one feather and no bacteria is also shown.

**Figure 5 marinedrugs-18-00537-f005:**
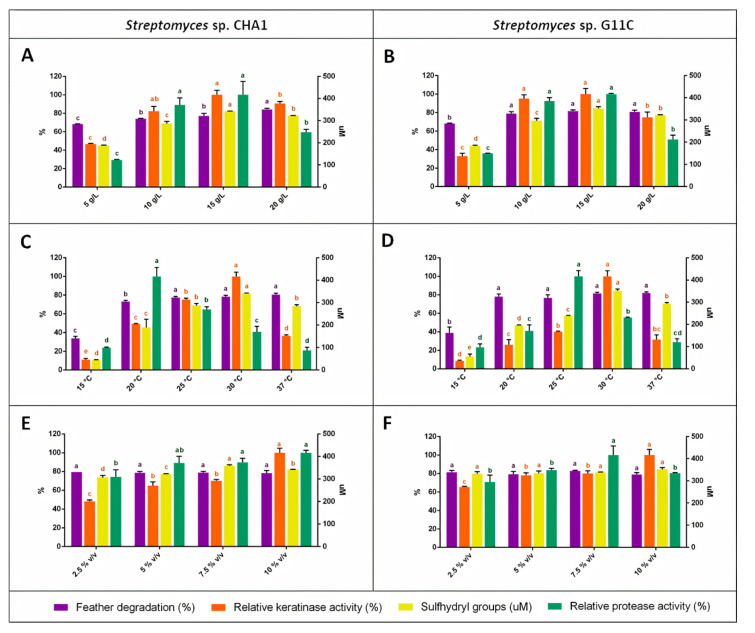
Factors affecting feather degradation, keratinolytic activity, sulfhydryl groups concentration, and proteolytic activity in the cultivation of *Streptomyces* sp. CHA1 and *Streptomyces* sp. G11C. (**A**,**B**) Effect of feather concentration (5, 10, 15, and 20 g/L) maintaining temperature (30 °C) and inoculum (10% *v*/*v*) constant; (**C**,**D**) Effect of temperature (15, 20, 25, 30, and 37 °C) maintaining substrate (15 g/L) and inoculum (10% *v*/*v*) constant; (**E**,**F**) Effect of inoculum percentage (2.5, 5, 7.5, and 10% *v*/*v*) on the cultivation of strains CHA1 and G11C maintaining temperature (30 °C) and substrate (15 g/L) constant, respectively.

**Figure 6 marinedrugs-18-00537-f006:**
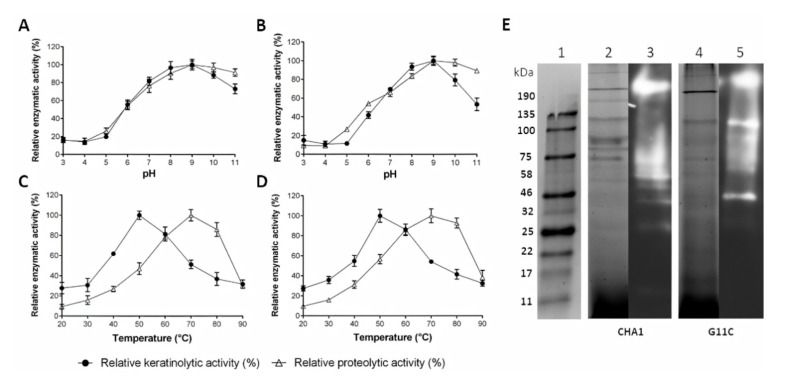
Partial characterization of the culture supernatants containing crude enzymes of *Streptomyces* sp. CHA1 and *Streptomyces* sp. G11C. (**A**) Effects of pH on the keratinolytic and proteolytic activity of the culture supernatant of *Streptomyces* sp. CHA1 and (**B**) *Streptomyces* sp. G11C; (**C**) Effects of temperature on the keratinolytic and proteolytic activity of the culture supernatant of *Streptomyces* sp. CHA1 and (**D**) *Streptomyces* sp. G11C. Values are represented as percentages of relative activity with respect to the maximum value obtained for each enzyme activity curve. (**E**) SDS-PAGE and zymography of the crude enzyme solutions of *Streptomyces* sp. CHA1 and *Streptomyces* sp. G11C. Lane 1: protein markers; lanes 2 and 3: SDS-PAGE and gelatin zymography of the crude enzyme solution of *Streptomyces* CHA1; lanes 4 and 5: SDS-PAGE and gelatin zymography of the crude enzyme solution of *Streptomyces* sp. G11C. The proteolytic activity can be visualized as a clear zone on the zymograms. Culture supernatants after five days of incubation, with turkey feathers as the sole source of carbon and energy, were used for all experiments. Crude enzyme solutions for SDS-PAGE and zymography were obtained by concentrating the culture supernatant 10-fold using an ultrafiltration process (with a membrane of 10 kDa molecular weight cut-off).

**Figure 7 marinedrugs-18-00537-f007:**
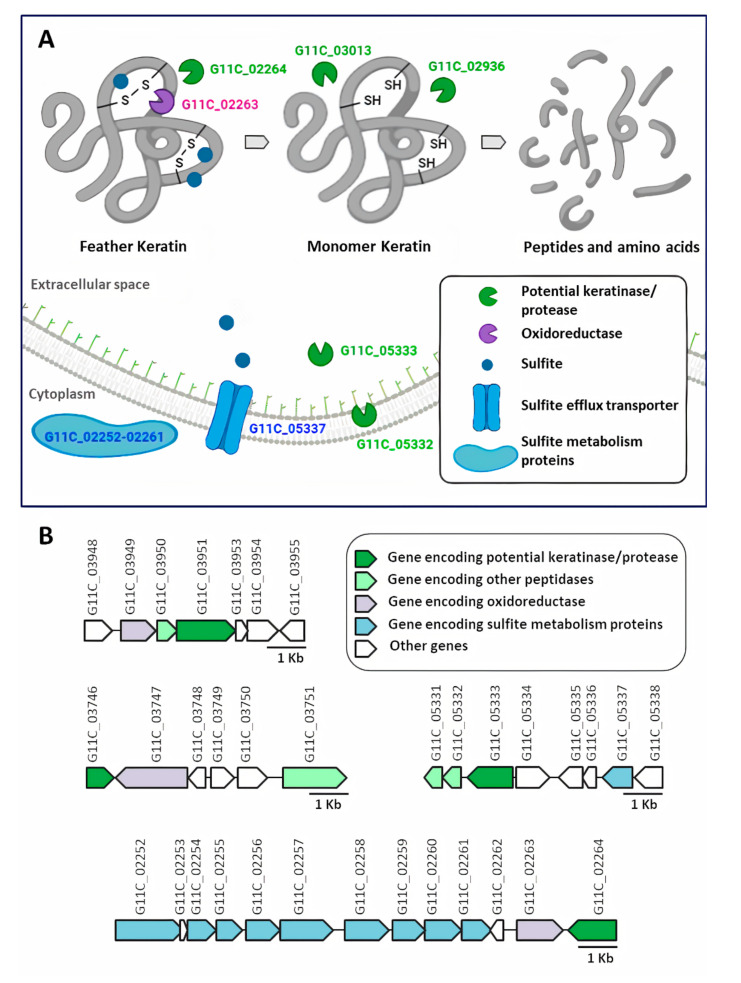
An illustrative possible model for the extracellular mechanism of keratin degradation in *Streptomyces* sp. G11C (**A**) and the genetic context of some secreted peptidases (**B**). Proteins are indicated with the number of their locus indicated in Table 2 and Appendix A and are represented by colors according to their function: green, potential keratinase/protease; purple, oxidoreductase; light blue, sulfite metabolism and transport. It should be noted that only the proteins represented in green color were identified in the proteomic analysis of the secretome of this strain. The other proteins were encoded by genes that were in the same genetic contexts as the green color proteins.

**Table 1 marinedrugs-18-00537-t001:** Maximum keratinolytic activity values for substrate, temperature, and inoculum parameters for *Streptomyces* sp. CHA1 and *Streptomyces* sp. G11C.

Strain	Activity	Substrate		Temperature		Inoculum	
		Max. Value	Parameter g/L	Max. Value	Parameter °C	Max. Value	Parameter % *v*/*v*
CHA1	Feather deg.	84.3 ± 1.0 *	20	79.1 ± 1.8	37	78.9 ± 1.3	7.5
	Keratinase act.	62.3 ± 3.0	15	63.5 ± 3.0 *	30	63.5 ± 3.0 *	10
	R-SH groups	342.4 ± 3.3 *	15	342.6 ± 4.2 *	30	358.4 ± 5.0 *	7.5
	Protease act.	11.85 ± 1.7	15	29.1 ± 2.8 *	20	13.9 ± 0.4	10
G11C	Feather deg.	81.7 ± 1.1	15	81.7 ± 1.1	30	82.8 ± 0.6	7.5
	Keratinase act.	78.3 ± 4.7	15	78.3 ± 4.7 *	30	78.3 ± 4.7 *	10
	R-SH groups	352.0 ± 7.5	15	352.0 ± 7.5 *	30	352.0 ± 7.5	10
	Protease act.	15.2 ± 0.1	15	27.4 ± 1.7 *	25	18.9 ± 1.8 *	7.5

Feather degradation values are presented in percentage, keratinolytic and proteolytic activities as U/mL, and sulfhydryl groups as µM. * Significant values compared to other measurements.

**Table 2 marinedrugs-18-00537-t002:** Selected proteases of the secretome of *Streptomyces* sp. G11C predicted in the bioinformatic analysis by Valencia et al. [40].

Seq-Illumina Locus	Description	MEROPS Classification	Molecular Function	Length	MolWt (MH)	Spectral Counts
G11C_03013	S8 family peptidase	s08	Serine endopeptidase	528	53,486.9	79
G11C_00267	Serine protease	s01	Serine endopeptidase	1427	152,855.9	60
G11C_05333	S1 family peptidase	s01	Serine endopeptidase	384	38,911.3	27
G11C_03247	M56 family metallopeptidase	m56	Metalloendopeptidase	311	32,205.8	18
G11C_03889	X-Pro dipeptidyl peptidase	s15	Aminopeptidase	597	62,112.5	15
G11C_02143	PrsW family intramembrane metalloprotease	m82	Metalloendopeptidase	318	34,048.5	12
G11C_05273	S8 family peptidase	s08	Serine endopeptidase	404	41,273.7	10
G11C_01742	Lon protease 2	s16	ATP-dependent serine endopeptidase	359	37,268.1	9
G11C_01512	S1 family peptidase	s01	Serine endopeptidase	360	36,229.1	9
G11C_03649	Hypothetical protein	nd		242	25,244.8	8
G11C_03746	PepSY domain-containing protein	nd	Peptidase propeptide	229	23,855.7	8
G11C_02264	Extracellular serine proteinase	s08	Serine endopeptidase	404	40,851.8	7
G11C_03951	Calpain family cysteine protease	c02	Cysteine endopeptidase	498	54,800.4	7
G11C_04177	S8_S53 family peptidase	s53	Acid-acting serine endopeptidase	456	47,500.6	7
G11C_02546	Trypsin-like serine protease	s01	Serine endopeptidase	296	29,365.6	6
G11C_03446	Class F sortase	nd	Sortase cysteine transpeptidase	140	13,909.3	5
G11C_02426	M50 family metallopeptidase	m50	Metalloendopeptidase	247	26,291.5	4
G11C_03494	Papain-like cysteine peptidase	nd	Cysteine endopeptidase/exopeptidase	219	23,970	3

MEROPS classification: s, serine peptidase; m, metallo peptidase; c, cysteine peptidase; nd, not determined.

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
