# Peer review of "Enzyme Bioprospection of Marine-Derived Actinobacteria from the Chilean Coast and New Insight in the Mechanism of Keratin Degradation in *Streptomyces* sp. G11C"

_marinedrugs, 2020, doi:10.3390/md18110537_

Round 1

Reviewer 1 Report

The MS by González et al. reports the bioprospecting of enzymes with biotechnological application from 75 marine-derived actinobacteria strains from the Chilean coast. Proteases, lipases, cellulases, amylases and keratinases were the enzymes mainly secreted by these selected strains. Streptomyces sp. CHA1 and Streptomyces sp. G11C resulted as the most interesting strains as keratinase producers. In particular, the crude extract of Streptomyces sp. G11C was chosen for a proteomic analysis to identify enzymes highly expressed, possibly involved in the mechanism of keratin degradation.

The content is appropriate for the journal Marine Drugs, and the work is interesting because it reports the importance of bioprospecting of marine microorganisms for industrial application. The manuscript is well written and well organized. However, to my opinion, the objective of the paper (the final result) was achieved only partially, because the authors analyzed the crude extract of the selected strain and not a pure active enzyme or at least an active fraction.

Moreover, since Streptomyces species have been recognized as one of the main producers of keratinases, it is not clear why in this strain the keratinolytic activity is exerted by 18 enzymes identified in the secretome of Streptomyces sp. G11C.  

My suggestion is to identify the most active fraction in the extract and work on it to analyze the composition of exact enzymes involved in the keratin degradation.

Author Response

Thank you very much for your comments. We agree with your suggestion of analyzing the most active fraction in the extract. Effectively, when proteins are identified from the crude extract, a wide variety of enzymes can be detected. In fact, in most studies only the main keratinase has been characterized, we cite as an example the work of Bressollier et al., 1999 (https://www.ncbi.nlm.nih.gov/pmc/articles/PMC91380/), who identified at least 6 extracellular proteases in the Streptomyces secretome. However, they only characterized the main enzyme with keratinolytic activity. However, when this approach is taken, it is not possible to determine other factors or enzymes, which may have an important role in the degradation mechanism.

Our first objective was to look at the full context of the secreted proteases to try to understand the mechanism of keratin degradation without leaving any important factor out, and later, as you mention, to identify the active fraction from the Streptomyces G11C secretome. Unfortunately, due to the global health crisis, our lab activities have been abruptly stopped since March, and we still don´t know when we open again. But, it is an objective that we will carry out at some point when activities are normalized. 

It is worth mentioning that considering our broader approach, we were able to identify a set of important factors to carry out the keratin degradation in this bacterium. First, the identification of various proteases, which belong to the S01 and S08 families, classifications in which most of the known keratinases have been reported. Second, the identification of a novel metalloprotease from the M06 peptidase family (the most abundant protein in the Streptomyces G11C secretome), which is the first time to be described in keratin degradation. Furthermore, the analysis of the genetic contexts of secreted proteases allowed us to identify adjacent genes encoding oxidoreductases and genes related to sulfite metabolism, factors that could be involved in disulfide bond breaking (see Results, lines 349 to 377 and Discussion, lines 516 to 540). Our broader approach permitted the identification of a set of keratinases that can be important for the complex mechanism of keratinolysis in strain G11C, and made it possible to propose a mechanism of degradation.

A revised manuscript has been uploaded, taking into consideration the comments of all reviewers.

Reviewer 2 Report

The authors characterized marine actinobacteria with their hydrolytic activities and clearly described the enzymes involved in the keratin degradation mechanism with disulphide reduction of Streptomyces sp. G11C. Most of the experimental approach and results are solid to support its main conclusion. This work would be worthy of publication but still there are some points that should be addressed.

  1. The authors have revealed and annotated the genome sequence in another manuscript. Have the authors found all the expected keratinases from the genome for the proteins revealed by the proteomics approach?
  2. Couldn’t the detailed role of each keratinase have been predicted by conservation of its amino acid sequence and domain content?
  3. In the present study, the authors identified keratinases that were expressed in media containing keratin, but were they not expressed in media without keratin?

Other minor points.

  1. Fig. 1, legend, “cellulose activity”: cellulase activity
  2. Line 327, Table S4: Table S5
  3. Line 339, 03949: 03649
  4. Line 339, hypothetical protein: How can the function of this protein be predicted from its amino acid sequence?
  5. Lines 581-583: Revise the subscript well.
  6. Lines 692-693: What is a “single-phase microcapillary column?” What is the column size?
  7. Line 694, “120-min single-phase step separation”: provide a few more details, such as flow rate, solvent contents, gradient profile, etc.

Author Response

Thank you for your helpful comments. Below, we address the points you numbered in your review:

1. The authors have revealed and annotated the genome sequence in another manuscript. Have the authors found all the expected keratinases from the genome for the proteins revealed by the proteomics approach?

Answer: Our previous work, Valencia et al. (currently under second revision), 7 putative keratinase sequences were identified with a bioinformatic integrative approach, in addition to another 17 genes belonging to orthologous groups of proteases, unique to the keratinolytic strain G11C. Of the 7 predicted putative keratinase sequences, 6 were found in the secretome, along with 12 of the unique proteases of the G11C strain. A more detailed description was added in lines 351-356, to clear this point.

2. Couldn’t the detailed role of each keratinase have been predicted by conservation of its amino acid sequence and domain content?

Answer: It still isn´t very clear how to concisely predict keratinases based on its amino acid sequence or domain content. There are no conserved amino acid signatures. Our previous work Valencia et al. (currently under second revision) addresses a possible prediction with the use of an integrative bioinformatic approach. 

To complement the role of the keratinases found in our study, the molecular function of each secreted protease selected according to Valencia et al. [40], was predicted. This prediction was made based on MEROPS classification (family and subfamily of proteases), domains present in the primary protein structure and presence of catalytic residues. It is observed that most of the selected proteases correspond to endopeptidases of the serine protease type. Presence of endopeptidases of the metalloprotease type and a cysteine ​​protease type are also observed. In addition, presence of exoproteases is also observed (aminopeptidase and cysteine exoprotease). This information revealing a more detailed role of the keratinases has been added in lines 375 to 379 and in an additional column inTable 2.

3. In the present study, the authors identified keratinases that were expressed in media containing keratin, but were they not expressed in media without keratin?

Answer: Effectively, it is a question that we have asked ourselves. In fact, experiments were under way to answer this question. Unfortunately, since March we were not able to continue work in the laboratories, and still do not know when our university will open again for normal activities.

Oher minor points.

  1. Fig. 1, legend, “cellulose activity”: cellulase activity. Has been changed accordingly.
  2. Line 327, Table S4: Table S5. Has been changed. Now in line 346.
  3. Line 339, 03949: 03649. Has been changed. Now in line 361.
  4. Line 339, hypothetical protein: How can the function of this protein be predicted from its amino acid sequence? 

Answer: According to our manual curation, the sequence of the G11C_03649 protein has neither conserved domains nor catalytic residues in its primary structure. It also does not present in the MEROPS classification. Therefore, it was annotated as a hypothetical protein. To clarify this point, a short phrase was added in lines 361-363 of the manuscript.

     5. Lines 581-583: Revise the subscript well.

Answer: Unfortunately, we don´t seem to have subscripts in those specific lines. Could you show us the text please?

     6. Lines 692-693: What is a “single-phase microcapillary column?” What is the column size?

Answer: Single-phase microcapillary column is a column that uses a single-phase step separation. To clarify it, we added a full description of the column and its preparation procedure as follows: “The resulting peptides were eluted in a 100 μm i.d. microcapillary column with a 5-μm pulled tip packed with 11 cm of 3-μm Aqua C18 resin (Phenomenex, USA), and directly analyzed in an LTQ‐Orbitrap XL mass spectrometer (Thermo Finnigan) using a 120‐min single‐phase step separation.” Please see lines 744 to 745.

      7. Line 694, “120-min single-phase step separation”: provide a few more details, such as flow rate, solvent contents, gradient profile, etc.

Answer: As requested, we added a complete description of LC methodology used: “The gradient profile consisting of 5 min in 100% (v/v) buffer A (5% acetonitrile/0.1% formic acid), 70 min in 15%, 30 min in 45%, 5 min in 75%, and 10 min in 100% (v/v) of a buffer B (80% acetonitrile/0.1% formic acid solution) was used. A cycle of one full-scan mass spectrum (300−2000 m/ z) followed by five data-dependent MS/MS spectra at a 35% normalized collision energy was repeated continuously throughout each step of the multidimensional separation. To prevent repetitive analysis, dynamic exclusion was enabled with a repeat count of 1, a repeat duration of 30 s, and an exclusion list size of 200. Application of mass spectrometer scan functions and HPLC solvent gradients were controlled by the Xcalibur data system (Thermo, USA).” See lines 747 to 754.

A revised manuscript has been uploaded, taking into consideration the comments of all reviewers.

Reviewer 3 Report

Manuscript "Enzyme bioprospection of marine-derived Actinobacteria ... "is devoted to the results of screening for marine actinomycetes in order to identify popular hydrolytic enzymes. In addition, the authors searched for producers of keratinolytic enzymes and demonstrated their effect on the example of two cultures. Of course, there is an obvious need for them, and new effective microorganisms capable of synthesizing these enzymes are required for biotechnology. In this sense, the relevance of this work is obvious. However, after reading the manuscript, I had several questions.

First of all, the structure of the text is puzzling. The work consists of two parts (screening of producers of hydrolytic enzymes and screening and characterization of the keratinolytic complex of the selected Streptomycetes), which, in my opinion, are not interrelated with each other. A detailed presentation of the screening results for cellulases, amylases, proteases and lipases in no way brings the reader closer to understanding the mode of action of keratinolytic enzymes. I think this part should be separated into a separate manuscript.

Second. In the text I did not find any mention of how the selected keratinolytic Streptomycetes were identified. It is necessary to provide a phylogenetic analysis and the identification of the selected microorganisms.

Third. Comparison of the effectiveness (specific activities, processing conditions, etc.) of the discussed microbial producers and their keratinolytic complex with known microbial producers and enzymes would greatly enhance the menuscript.

Fourth. The authors mentioned crude keratinase preparations, which they compared with each other. Meanwhile, this is only a culture liquid after certain periods of growth. A crude preparation means at least one purification step (eg sulfate precipitation, desalting, concentration). This was not done and it is unreasonable to compare the data in such a way due to numerous factors affecting the final results.

Author Response

Thank you for your helpful comments. Below, we address the points you numbered in your review:

First of all, the structure of the text is puzzling. The work consists of two parts (screening of producers of hydrolytic enzymes and screening and characterization of the keratinolytic complex of the selected Streptomycetes), which, in my opinion, are not interrelated with each other. A detailed presentation of the screening results for cellulases, amylases, proteases and lipases in no way brings the reader closer to understanding the mode of action of keratinolytic enzymes. I think this part should be separated into a separate manuscript.

Answer: Thank you for your suggestion, indeed as you mentioned, the manuscript covers two topics (screening of extracellular enzymes and understanding of the mechanism of keratin degradation by one of the selected strains). We had also thought about your suggestion, but since the special Issue of the Marine Drugs Journal is focused on Marine Enzymes, it seemed like a good opportunity to publish all our work. Taking into consideration the opinion of the other two reviewers, we have decided to leave both parts of the paper. Anyway, we warmly appreciate your suggestion.

Second. In the text I did not find any mention of how the selected keratinolytic Streptomycetes were identified. It is necessary to provide a phylogenetic analysis and the identification of the selected microorganisms.

Answer: The streptomycete strains we enzymatically tested for this manuscript were isolated and identified in our previous papers: 

Claverías et al 2015: https://www.frontiersin.org/articles/10.3389/fmicb.2015.00737/full 

Undabarrena et al 2016: https://www.frontiersin.org/articles/10.3389/fmicb.2016.01135/full  

Cumsille et al 2017: 

https://www.ncbi.nlm.nih.gov/pmc/articles/PMC5618425/pdf/marinedrugs-15-00286.pdf

To improve the manuscript, additional information was added to enhance this information (lines 91-92). We also mention the isolation of these strains in lines 110-111, 563-565.

Considering that in Cumsille et al (Fig. 3), we have a phylogenetic tree with our streptomycetes, including strains G11C and CHA1, we decided not to add a phylogenetic tree in this manuscript.

Third. Comparison of the effectiveness (specific activities, processing conditions, etc.) of the discussed microbial producers and their keratinolytic complex with known microbial producers and enzymes would greatly enhance the manuscript.

Answer: We agree with your suggestion. We were very interested in obtaining a known microbial producer to compare with our strains. But the cost of this commercial microorganism was excessively high and out of our budget. With respect to comparing with enzymes, during our study we decided not to compare with enzymes at this stage of the work until we obtained a purified fraction. Certainly, these are the following steps that we plan to accomplish in the future.

Fourth. The authors mentioned crude keratinase preparations, which they compared with each other. Meanwhile, this is only a culture liquid after certain periods of growth. A crude preparation means at least one purification step (eg sulfate precipitation, desalting, concentration). This was not done and it is unreasonable to compare the data in such a way due to numerous factors affecting the final results.

Answer: We appreciate your feedback and made changes to improve our manuscript, considering your suggestions. Generally, in various screening works accomplished for enzymatic activity comparisons, using different strains or culture conditions, culture supernatant is used. We cite some examples in this regard: Nnolim et al., 2020: https://www.sciencedirect.com/science/article/pii/S2215017X20300497; Barman et al., 2017: https://www.ncbi.nlm.nih.gov/pmc/articles/PMC5608654/.

We agree that comparing and characterizing enzymes of interest requires at least one purification step. In our work, we show the keratinolytic activity screening in various marine actinobacterial strains and characterize various culture conditions in two streptomycete strains. In these stages, we use the culture supernatant, which we call “enzyme crude solution”. Subsequently, when characterizing the secretome of the strains (SDS-PAGE gel and proteomics analysis), the supernatants were concentrated through an ultrafiltration process (more information in lines 712-713). Therefore, considering your comment, and to avoid misunderstandings, "enzyme crude solution" has been changed to "culture supernatant" (see lines: 100, 293-294, 320-324 (throughout Figure 6 legend), 643, 644, 659, 671, 700, 710, 779). The term "enzyme crude solution" will only be used when we describe the analysis of protein gels (Figure 6 legend: lines 320-324), where there was at least one purification step.

Round 2

Reviewer 1 Report

The authors have replied to all my comments satisfactorily, the manuscript is now improved. I feel satisfied of this revision, so I support the publication of the manuscript on the journal Marine Drugs.

Reviewer 3 Report

I cannot say I am completely satisfied with the answers. The references to other reviewers and to the topic of the Special issue of the Journal, the reply about a lack of opportunities to carry out experiments seem especially fanny. Nevertheless, some of my comments were accepted and corrections were made. I would like to wish the Authors good luck.